# The Multifaceted Bacterial Cysteine Desulfurases: From Metabolism to Pathogenesis

**DOI:** 10.3390/antiox10070997

**Published:** 2021-06-23

**Authors:** Mayashree Das, Arshiya Dewan, Somnath Shee, Amit Singh

**Affiliations:** Centre for Infectious Disease Research, Department of Microbiology and Cell Biology, Indian Institute of Science, Bangalore 560012, India; mayashreedas@iisc.ac.in (M.D.); arshiyadewan@alum.iisc.ac.in (A.D.); somenathshee@iisc.ac.in (S.S.)

**Keywords:** cysteine desulfurase (CSD), Fe-S cluster, redox, thio-cofactors, ROS

## Abstract

Living cells have developed a relay system to efficiently transfer sulfur (S) from cysteine to various thio-cofactors (iron-sulfur (Fe-S) clusters, thiamine, molybdopterin, lipoic acid, and biotin) and thiolated tRNA. The presence of such a transit route involves multiple protein components that allow the flux of S to be precisely regulated as a function of environmental cues to avoid the unnecessary accumulation of toxic concentrations of soluble sulfide (S^2−^). The first enzyme in this relay system is cysteine desulfurase (CSD). CSD catalyzes the release of sulfane S from L-cysteine by converting it to L-alanine by forming an enzyme-linked persulfide intermediate on its conserved cysteine residue. The persulfide S is then transferred to diverse acceptor proteins for its incorporation into the thio-cofactors. The thio-cofactor binding-proteins participate in essential and diverse cellular processes, including DNA repair, respiration, intermediary metabolism, gene regulation, and redox sensing. Additionally, CSD modulates pathogenesis, antibiotic susceptibility, metabolism, and survival of several pathogenic microbes within their hosts. In this review, we aim to comprehensively illustrate the impact of CSD on bacterial core metabolic processes and its requirement to combat redox stresses and antibiotics. Targeting CSD in human pathogens can be a potential therapy for better treatment outcomes.

## 1. Introduction

Sulfur (S) is an essential element for life [1]. Biomolecules contain S predominantly in amino acids such as methionine and cysteine or thio-cofactors and thio-nucleosides. Cysteine can be accessed via inorganic sources (sulfate (SO_4_^2−^), sulfide (S^2−^), thiosulfate (S_2_O_3_^2^^−^), and sulfonate (R-SO_3_^−^)), reverse transsulfuration pathway, glutathione, and cysteine import [2]. A multi-protein S-relay system then utilizes intracellular cysteine to synthesize thio-cofactors and thio-nucleosides [1]. Since excess of soluble-S (S^2^^−^/cysteine) induces metabolic paralysis, the S-relay pathway imparts significant evolutionary advantages by providing an alternative route of S transfer to essential thio-cofactor binding proteins without allowing soluble S to reach toxic levels [3].

S is mobilized from cysteine to diverse biomolecules via cysteine desulfurase (CSD). Cysteine desulfurases (CSDs: plural) are ubiquitous and highly conserved PLP (pyridoxal-5′-phosphate)-dependent enzymes [4]. They decompose L-cysteine to L-alanine while releasing sulfane S (S^0^) via the formation of an enzyme-linked persulfide (R-SSH(R)) intermediate on its conserved cysteine residue [5]. The persulfide S formed is then utilized to generate wide-ranging S-containing cofactors involved in various biosynthetic pathways [4,5,6]. This S-trafficking enzyme was first discovered in *Azotobacter vinelandii* as part of a gene cluster involved in nitrogen fixation (*nif*) [7]. This CSD named NifS catalyzes the formation of Fe-S clusters specifically in the nitrogenase enzyme complex by providing inorganic sulfide [7]. Following this, a homolog of NifS was characterized for its role in Fe-S cluster biogenesis of housekeeping proteins and named IscS in *A. vinelandii* [8]. Subsequently, several CSDs across various kingdoms were characterized and termed homologs of NifS, IscS, and SufS [9,10,11]. CSD is pivotal in Fe-S cluster assembly, tRNA modification, and biosynthesis of thiamine, biotin, lipoic acid, molybdopterin, nicotinamide adenine dinucleotide (NAD), and branched-chain amino acids [4,5]. In addition to its function in primary metabolism, CSD expressed by soil-dwelling *Nocardia* and *Streptomyces spp.* is involved in donating S to form thiotetronate-ring of an antimicrobial metabolite, thiolactomycin (TLM) [12,13,14]. Studies have shown anti-mycobacterial, anti-malarial, and anti-trypanosomal activities of TLM [15,16,17,18]. The maintenance of intracellular redox homeostasis is imperative for the survival of pathogens [19]. Within their host, pathogens face an onslaught of oxidative and nitrosative stress [19]. Remarkably, CSD has been shown to protect bacteria from oxidative stress generated within the phagocytes and by antibiotics, plus chemical agents such as potassium tellurite or oxidants (menadione, plumbagin, and cumene hydroperoxide) [20,21,22,23,24]. In sum, CSD contributes to not only housekeeping functions but also mediates stress-defense and pathogenicity.

The first review on CSD was published in 2002 by Mihara et al. [4]. In the following two decades, there has been tremendous progress on understanding the mechanism and function of CSD in diverse organisms, including major human pathogens. This paper aims to comprehensively summarize and underline CSD as a pivotal node linking its essentiality in diverse biochemical pathways with evasion from oxidative damage induced by the host immune pressures to antibiotics and exogenous chemicals. Therefore, CSD exhibits all criteria for a potential drug target and demands the discovery of inhibitor molecules to combat the rise in antimicrobial resistance (AMR).

## 2. Classification and Distribution of Cysteine Desulfurases

### 2.1. Classification

CSD (EC 2.8.1.7) belongs to the class of sulfur-transferases and catalyzes the transfer of S from substrate L-cysteine to various partner carrier proteins to synthesize S-containing cofactors and thio-nucleotides [4,5,6]. The first discovered CSD was NifS (Nitrogen-Fixation) from *A. vinelandii* [11,25]. NifS is involved in the maturation of the metallocluster (Fe-S) of the nitrogenase complex in nitrogen-fixing bacteria, certain microaerophiles, and non-nitrogen-fixing anaerobes [5,7,26]. Subsequently, enzymes belonging to this family (IscS, SufS, CsdA, YrvO, and DndA) were discovered in organisms across bacterial species [4,27,28,29,30]. These CSD homologs differ at the level of primary amino acid sequence, local structural organization, and enzyme kinetics, based on which CSDs are classified into two categories (Table 1) [4,5]. Members of Class I CSDs are similar to NifS and IscS, whereas Class II includes SufS-like and cysteine sulfinate desulfinase A (CsdA)-like proteins (Figure 1A) [4,5]. Sequence alignment of CSDs delineates two primary differences—a 12-residue sequence insertion after the active site cysteine in Class I CSDs (Figure 1B). In contrast, Class II members contain a distinct sequence insertion near the PLP-coordinating Lysine [4,5,6]. The consensus sequences around the conserved cysteine also act as an identifier -SSGSACTS- in Class I, whereas -RXGHHCA- in Class II [5]. Furthermore, the catalytic loop containing the nucleophilic cysteine is extended and structurally more flexible in Class I enzymes than Class II enzymes [4,6]. Additionally, the participation of Class I and Class II proteins in distinct pathways is influenced by the interacting protein partners. For example, the IscS (Type I) enzyme participates in Fe-S cluster biogenesis upon interaction with the S acceptor (IscU), whereas its participation in (thio-)tRNA biosynthesis is mediated via interaction with TusA and Thil [31,32,33]. The multiple interacting partners of the Class I CSDs are likely due to the flexible catalytic loop; however, NifS and *Bacillus subtilis* NifZ interact with their specific partners NifU and ThiI, respectively [34,35]. Unlike Type I CSDs, Type II CSDs such as SufS [36,37] and CsdA [38] interact with specific S acceptor proteins SufE or SufU and CsdE, respectively, for Fe-S cluster biogenesis and in the generation of cyclic threonylcarbamoyladenosine (ct^6^A) at 37th position (for CsdA). N^6^ct^6^A and its derivatives in tRNA are responsible for ANN codon usage [39]. Due to structurally defined shorter catalytic loop, Type II CSDs’ activity is exclusively dependent on its interaction with specific partner proteins. Another class of CSD has been reported in cyanobacteria *Synechocystis* PCC 6714. This PLP-dependent L-cyst(e)ine C-S-lyase (C-DES) is a monomer and lacks a conserved cysteine residue at the active site, explaining its insensitivity to alkylating agents [40,41]. It generates sulfide, ammonia, and pyruvate from cysteine/cystine instead of alanine and sulfane S [40,41]. The crystal structure and in vitro studies suggest the formation of a product-cysteine-persulfide in C-DES processed reaction compared to enzyme-linked persulfide generated by the other classes of CSD [40,41]. C-DES has been shown to participate in the Fe-S cluster biogenesis of ferredoxin [40,41].

### 2.2. Distribution

CSDs exist in all three domains of life. The genes encoding for NifS, IscS, SufS, and CsdA were discovered within the Fe-S cluster biosynthetic machinery—NIF, ISC, and SUF (SUF-like) [43]. The genes involved in the Fe-S cluster biosynthesis are arranged in operons in most bacteria, while there are exceptions of dispersed genes (Figure 2) [43,44]. The Fe-S biogenesis systems in *Escherichia coli* and *A. vinelandii* are the most extensively studied. *E. coli* contains both the ISC and the SUF systems, while *A. vinelandii* contains the NIF and ISC systems [45,46]. Comparison of bacterial genomes revealed that microorganisms differ in the number and type of operons they utilize. For instance, most Gram-positive bacteria, archaea, and cyanobacteria contain only a *suf* operon and display specific differences from the *E. coli* counterpart. Compared *to E. coli*, the SUF system in Gram-positive bacteria codes for an IscU-like scaffold protein in place of SufU and lacks S-transfer protein SufE [45]. In addition to SufS, the model Gram-positive bacterium *Bacillus subtilis* harbor three other CSDs: NifZ, NifS, and YrvO. Moreover, the human pathogen *Mycobacterium tuberculosis (Mtb)* consists of only a *suf* operon (*sufRBDCSUT*) with two additional genes, *sufR* a transcriptional regulator [47,48] and *sufT* involved in Fe-S cluster maturation [49,50], in addition to a single gene of the *isc* operon, *iscS* [51,52].

## 3. Structure and Reaction Mechanism of Cysteine Desulfurase

### 3.1. Structure

The crystal structure of both Type I (*Thermotoga maritima* NifS, *Mtb* IscS, and *E. coli* IscS) [51,53,54] and Type II CSDs (*E. coli* CsdB/SufS, *B. subtilis* SufS, and *Synechocystis sp. PCC 6803* SufS) [55,56,57] have been solved. Based on amino acid sequence and three-dimensional structural analysis, CSDs are grouped into the fold-Type I family of PLP-dependent enzymes and belong to the aminotransferase class V sub-family (Figure 3A) [6,58,59]. These enzymes form a homodimer with a structural distinction into a small N- and a large C-terminal domain. Each monomer contains a PLP cofactor, bound covalently to a conserved lysine residue (Lys206 in *E. coli* IscS) in the active site pocket near the dimer interface. The secondary structures building the CSD comprise both α-helices and β-sheets, with α-helices making the major fraction (40.1% in *E. coli* IscS) [54]. A distinct loop extending between the small and large domain contains catalytic cysteine residue (Cys328 in *E. coli* IscS) and forms one side of the entrance-channel for the substrate towards the CSD active site. As discussed above, in most Type I enzymes, this catalytic loop is extended (Figure 3B). The catalytic cysteine is located far away from active-site pocket (>1.7 nm in *E. coli* IscS), suggesting that the steps of desulfuration are sequential and not cooperative [54]. However, the loop, being structurally disordered, is highly flexible and undergoes a large conformational change in every catalytic cycle [4,6,60]. This catalytic loop movement is critical since a mutation adjacent to the loop (Ala 327Val in *Salmonella* IscS) probably restricts this movement by β-branching of Val residue. This inhibits IscS catalysis and thereby induces thiamine auxotrophy and decreased level of thionucleosides [61]. On the other hand, the catalytic loop is shorter and more rigid for the Type II enzymes and located near the active site (∼0.7 nm in *E. coli* SufS) (Figure 3C) [6,60]. Type II enzymes also harbor a characteristic insertion-sequence near the PLP-binding lysine residue, which is absent in the IscS-like enzymes. This region forms a β-hairpin structure that likely imparts structural rigidity to the catalytic loop [54,60]. These structural properties can answer specific characteristics of Type II enzymes: (i) decreased length and flexibility of catalytic loop coerce CSD to depend on partner proteins for attaining favorable reaction kinetics; and (ii) mutations in CSD that alter the topology of this β-hairpin structure lead to the weak interaction between CSD and interacting partner protein, suggesting that β-hairpin structure probably evolved to improve interaction with the partner proteins [62].

### 3.2. Reaction Mechanism

CSD performs the desulfuration of its substrate L-cysteine, forming L-alanine and S^0^ or S^2−^ (under reducing conditions). The aldehyde group of the PLP cofactor allows the formation of an imine with a free amino group, and the pyrimidine ring structure acts as an electron-sink which are crucial interactions for the reaction to occur [62,63]. CSD’s reaction cascade can be divided into two sequential stages: (i) formation of the enzyme-linked cysteine persulfide (R-S-S-SH); and (ii) the transfer of the persulfide to respective S-acceptor proteins [4,26,57].

Persulfide formation on the enzyme occurs through the formation of following intermediate states (Figure 4): (i) transition of the internal Lys-aldimine (resting state) to the external Cys-aldimine (Schiff base), which requires sequential proton transfers and via formation of a geminal aldimine; (ii) the external aldimine formed creates an electronic coupling of the imine and pyrimidine ring of the PLP, generating a conjugated pi-electron withdrawal system, which aids in proton abstraction from substrate cysteine by a basic residue (probably His104 in *E. coli* IscS [6,26,64]) acting as a general base; (iii) it is followed with the active site Cys-thiol acting as a general acid and donates a proton to form Cys-PLP quinonoid adduct; (iv) this paves the way for the reaction for the nucleophilic attack on the substrate-thiol by the conserved cysteine, resulting in the formation of the enzyme-linked persulfide intermediate and an alanine-enamine PLP adduct; (v) general acid/base reactions lead to formation of Ala-ketimine and then Ala-aldimine, probably mediated by CSD’s His-residue; and (vi) finally, through reverse order of substrate binding steps, the product alanine is released with restoration of Lys-PLP internal aldimine [6].

The next step involves the transfer of terminal S to cysteine residues in acceptor proteins via persulfide intermediates. This is proposed to occur either by a nucleophilic attack on S acceptor by the enzyme’s persulfide S or vice versa [65]. The kinetic analysis demonstrates a ping-pong mechanism where product alanine is released first, followed by transpersulfuration to partner proteins [37]. However, in the absence of acceptor proteins, the persulfide can have two different fates based on the pertaining redox environment: (i) persulfide is released as S^2−^/H_2_S under reducing conditions; or (ii) enzyme linked-polysulfide species S_n_; (2 < n < 7) are formed on an adjacent free thiol under non-reducing conditions [66]. This process of persulfide transfer to the acceptor proteins is a significant advantage as it ensures the safe mobilization of S in a nontoxic form (sulfides) [3,67,68,69]. The mechanism of persulfide transfer and type of interacting S-acceptor partners differ among the two classes as described above. In vitro and in vivo studies in *E. coli* and *B. subtilis* have highlighted the phenomenon of enhanced desulfurase activity of SufS (Type II) in a tight complex-formation with SufE/SufU (by ∼30–50 fold [70]), which is not in the case for IscS/IscU interactions [37,70]. SufE also interacts with the scaffold component SufB of the SufB_2_C_2_/SufBC_2_D complex, which further enhances the activity of SufS by accelerating the transpersulfuration process by ∼100 fold [70]. This requirement for partner proteins by CSD for optimal activity suggests that CSD catalysis can be targeted by inhibiting the enzyme directly and by impeding the availability or function of S-acceptor proteins. Contrary to the specific interaction of SufS-SufE, SufS-SufU, and CsdA-CsdE, IscS can interact with multiple S-carriers like IscU, ThiI, and TusA. However, the requirement of partner/acceptor proteins for IscS expressed by the human pathogen *Mtb*, is not properly understood because *Mtb* contains only the CSD IscS and lacks other components of the ISC system. Despite this, *Mtb* IscS physically interacts and assembles Fe-S cluster into *Mtb* Fe-S proteins such as WhiB3, aconitase (Acn), and succinate dehydrogenase (SdhB). [51,71]. Therefore, the following questions need investigation to understand *Mtb* IscS activity comprehensively:(i)What is the mechanism of IscS mediated S-relay in *Mtb*?(ii)What scaffold protein participates in this process since IscS do not physically interact with *Mtb* SufU [51]?(iii)Is IscS alone sufficient to build Fe-S clusters?(iv)Does IscS contribute to stress tolerance and pathogenesis in *Mtb*?

## 4. CSD Controls Basal Metabolism by Mobilizing S from Cysteine to Diverse Cellular Pathways

Several studies indicate that IscS and CsdA contribute to sustain housekeeping function by participating in Fe–S biogenesis, thiamine or molybdopterin biosynthesis, and tRNA modification, as described in detail below (Figure 5). In contrast, SufS activity generally restricts to Fe–S cluster biogenesis under stress conditions such as oxidative stress, nitrosative stress, and Fe-limitation [1,5,6].

### 4.1. Fe-S Cluster Assembly

Fe-S clusters are susceptible to oxidation by molecular oxygen (O_2_) and its reactive intermediates (reactive oxygen species (ROS): hydrogen peroxide (H_2_O_2_) and superoxide (O_2_^•−^)), which leads to enzyme-inactivation and disruption of cellular physiology. More specifically, Fe^2+^ leached from oxidatively damaged Fe-S clusters can trigger the Fenton reaction to generate highly oxidative hydroxyl radicals (^•^OH), which oxidizes biomolecules at near diffusion-limited rates [44,72]. In addition to this, free S overload gives rise to toxic polysulfides inside cells [3]. Therefore, to abate deleterious consequences of free Fe and S, organisms have evolutionarily acquired multicomponent systems for calibrating Fe-S cluster biogenesis [43,44,72]. The regulation and biogenesis of Fe-S clusters have been extensively reviewed, and readers are encouraged to refer to these reviews [43,73,74]. Briefly, Fe-S cluster biogenesis encompasses two key steps: (i) assembly of the Fe-S cluster on the scaffold protein; and (ii) Fe-S cluster trafficking from scaffolds to the respective apoprotein targets. The starting point in Fe-S cluster biogenesis is the acquisition of the elemental components—Fe and S, where cysteine acts as the S source. The protein(s) involved in donating Fe for Fe-S cluster biogenesis remains elusive and needs further research. Subsequently, Fe-S clusters are assembled on the scaffold protein(s) via interaction with S and Fe donating protein complexes. Finally, the Fe-S clusters are catered to apoproteins with the help of carrier proteins. *E. coli* ISC system comprises IscR (a transcriptional regulator), IscS (a CSD), IscU (scaffold protein), IscA (an A-type carrier protein), HscB (a DnaJ-like co-chaperone), HscA (a DnaK -like chaperone), Fdx (a ferredoxin), and IscX [75]. The SUF system expresses SufR (a transcriptional regulator), SufS (a CSD), SufE/SufU (enhances SufS activity), SufBCD (scaffold complex), SufA (an A-type carrier protein), and SufT (accessory protein for Fe-S cluster maturation) [49]. The NIF system comprises the cysteine desulfurase NifS and the scaffold protein NifU [11].

The significant contribution of CSD is towards the formation of the Fe-S clusters, which act as cofactors to multiple enzymes. The tricarboxylic acid (TCA) cycle enzymes such as Acn and succinate dehydrogenase are Fe-S cluster-containing proteins [43]. Additionally, the auxotrophy of *E. coli iscS* mutant observed for nicotinic acid, isoleucine, and valine is consistent with the presence of Fe-S cluster on enzymes involved in their biosynthesis [76]. For instance, the precursor of NAD, quinolinic acid, is synthesized by a Fe-S cluster protein- quinolinate synthase A, whereas another Fe-S protein, dihydroxy acid dehydratase, is involved in the biosynthetic pathways of isoleucine and valine [4]. Fe-S cluster proteins also function as sensors (FNR, SoxR, IscR, SufR, WhiBs, etc.) to environmental cues that include gases (O_2_ and nitic oxide (NO)), Fe-starvation, ROS, reactive nitrogen intermediates (RNI), and redox-active compounds, which are relayed downstream for the regulation of gene expression and stress tolerance [77,78].

In *E. coli* and *Mtb*, the deletion of *iscS* reduced growth and lower activity of Fe-S cluster-dependent enzymes (2–50% of wild-type) [51,76]. Disruption of *iscS* is not lethal under standard growing conditions, indicating that alternate CSD-SufS could compensate for the loss of *iscS* in *E. coli* and *Mtb*. Consistent with this, overexpression of SufS restores active growth and enzymatic activity of Fe-S cluster proteins in the IscS mutant of *E. coli* [79]. The overlapping role of CSD-*sufS* and *iscS* is evident by the inability of *E. coli* to survive upon disruption of both the genes [79]. Taken together, these genetic studies demonstrate that CSDs are indispensable for bacterial metabolism and growth.

### 4.2. tRNA Modification (Thiolation)

Thio-modifications of transfer RNAs (tRNA) such as 2-thiouridine (s^2^U), 4-thiouridine (s^4^U), 2-thiocytidine (s^2^C), 2-methylthioadenosine (ms^2^A), and 2-thiouridothymidine (s^2^T54) are commonly observed across kingdom [80,81]. These thio-modifications are on seven tRNA positions: (5′-8, 9, 32, 33, 34, 37, and 54-3′). Except for thiolated adenines, thionucleosides are mostly formed by replacing the keto-oxygen of the nucleotide base with S [9,65,82,83,84,85]. Importantly, the positions at which thio-modifications occur are crucial for tRNA function and topology. Modification on the anticodon stem-loop aids in translational accuracy, whereas modifications at the T-loop and D-loop maintain structural stability and serve as a recognition motif for the tRNA aminoacyl synthetase and tRNA-modifying enzymes [80]. Remarkably, the s^4^U8 tRNA-modification positioned at the acceptor-arm and D-arm junction acts as a near-UV light photosensor and is highly conserved among bacteria [32]. Mechanistically, near-UV light crosslinks s^4^U with cytidine 13, which induces structural disorder, halts aminoacylation, and impedes translation to signal activation of stringent response in bacteria [32]. Interestingly, hypoxia-induced persistence of *Mycobacterium bovis* BCG is associated with fluctuations in the levels of modified tRNA and reduction in translation efficiency [86].

Generally, IscS provides S for thio-modifications, except s^2^U modification in *B. subtilis,* where YrvO functions as a CSD [80,87]. After S release by CSD, diverse S-carrier proteins facilitate S- transfer to the modification enzymes. The s^4^U modification involves ThiI protein that contains the PP-loop motif (ATP- binding) and activates the C4 atom of tRNA at U8 via adenylated intermediate. The activated tRNA is next thiolated by the ThiI persulfide derived from the action of IscS [88,89,90,91]. Contrary to this, s^2^U modification at the wobble position 34 is tailored by the enzyme MnmA (also known as TrmU), with IscS as the S donor and the TusABCDE system working as the S transfer mediators [81,89]. Intriguingly, *Mtb trmU* is proposed to be in an operon with *iscS* (the only gene of the ISC component in *Mtb*) [51]. The significance of such operonic arrangement on *Mtb* pathogenesis is unknown and needs to be elucidated to understand the effect of CSD on *Mtb* tRNA levels. Similar to ThiI, MnmA also consists of a PP-loop domain required for the activation of the C2 atom of U34 by adenylation. On the other hand, the s^2^C32 modification requires the involvement of a [4Fe-4S] cluster containing protein TtcA in which the Fe-S cluster is sufficed by the IscS-IscU system [65,92,93]. The formation of methylthio-derivatives (ms^2^A37) is catalyzed by methylthiotransferases MiaB and MtaB, both belonging to the radical S-adenosyl-L-methionine (SAM) superfamily, which requires Fe-S clusters for activity [9,83,84,85]. Hence, the pathways for the synthesis of thiolated tRNA can be segregated based on their dependence on Fe-S cluster-containing proteins [80].

Direct evidence of the involvement of the CSD IscS in tRNA thiolation is shown in *Salmonella enterica* serovar Typhimurium, where a strain defective in IscS function exhibited reduced levels of all five thiolated nucleosides present in tRNA [61]. Another study on *E. coli* IscS showed downregulation of two enzymes from the pyrimidine salvage pathway; namely, uridine phosphorylase and cytidine deaminase in the IscS deleted strain, suggesting diminished nucleotide metabolism and, therefore, reduced tRNA-thio-modifications [94]. Additionally, in vitro studies have shown the involvement of CSDs in the incorporation of selenium (Se) into Se containing proteins and tRNA modifications [95,96]. Indeed, *E. coli iscS* deleted strain were inept in synthesizing 5-methylaminomethyl-2-selenouridine in the wobble position of glutamate, lysine, glutamine- tRNA, and suffered a reduction in the levels of selenium-containing enzyme formate dehydrogenase H (FdhH) [96]. In sum, CSD has a dual function in S and Se mobilization.

### 4.3. Lipoic Acid Synthesis

Lipoic acid is an essential organosulfur cofactor for enzymes carrying out oxidative and one-carbon metabolism [97,98]. Lipoate remains covalently attached to the E2 subunit of these complexes through an amide linkage at a conserved N-terminal lysine residue [97,98,99]. In *E. coli*, lipoate synthesis is carried out by two reactions: (i) transferring an octanoyl group from octanoyl acyl carrier protein (octanoyl-ACP) to the apoprotein catalyzed by LipB (octanoyl transferase); and (ii) insertion of two S atoms at the C6 and C8 positions of octanoyl chain forming the dithiolane ring of lipoate by LipA (lipoate synthase) [97,98,100]. Additionally, cells can acquire free lipoate by scavenging, which is mediated by the enzyme lipoate-protein ligase A, LplA by a two-step ATP-dependent reaction to form protein conjugated lipoate. LipA belongs to the radical SAM -superfamily proteins harboring two [4Fe-4S]^+^ clusters, where one of the clusters is coordinated with CX_4_CX_2_C motif, common to all radical SAM enzymes. The second Fe-S cluster is coordinated by the cysteines of CX_4_CX_5_C motif, found only in lipoyl synthases [98,100,101]. Lipoate synthesis proceeds through an extraordinary self-sacrifice reaction step where S is derived from an auxiliary [4Fe-4S] cluster of LipA, which gets degraded after a single catalytic turnover [101,102]. CSD (IscS), NfuA, and IscU (Fe-S biogenesis and repair pathway proteins) efficiently reconstitute these clusters with non-rate-limiting kinetics and prime LipA for the next round of catalysis [101,102,103]. Therefore, CSD is essential for lipoate biogenesis.

The lipolyated enzymes are pyruvate dehydrogenase (PDH), α-ketoglutarate dehydrogenase (KGD), branched-chain keto acid dehydrogenase (BCKDH), acetoin dehydrogenase, and glycine cleavage system [97,98,104]. PDH and KGD participate in central carbon metabolism, while BCKDH functions to degrade branched-chain amino acids to generate branched-chain coenzyme A (BC-CoA), which can feed into the TCA cycle or anabolized for branched-chain fatty acid synthesis [98,99]. Interestingly, clinical data from patients with multi-drug resistant tuberculosis suggest an upregulation of the *Mtb lipB* [105]. Deletion of *dlaT* (the E2 of PDH and PNR/P) had a pronounced effect on *Mtb*’s growth in a standard medium in vitro, sensitized *Mtb* to RNI, and attenuated *Mtb* in the mouse [106]. Additionally, species-specific DlaT inhibitors selectively killed non-replicating *Mtb* [107]. Deletion of E3 subunit of PDH that completes the catalytic cycle of E2 also leads to a severe defect in bacterial growth inside the host [108].

### 4.4. Biotin Biogenesis

Biotin (vitamin B7) is an essential micronutrient required for amino acid metabolism, fatty acid biosynthesis, and replenishment of TCA cycle intermediates [109]. The known biotin-dependent enzymes, such as pyruvate carboxylase and acyl-CoA carboxylases (ACCs), utilize biotin-cofactor to transfer carbon dioxide between metabolites in carboxylation, decarboxylation, and trans-carboxylation reactions [109]. *Mtb* employs the ACCs to convert substrates such as acetyl-CoA, propionyl-CoA, and butyryl-CoA into intermediates for fatty acid and polyketide biosynthesis, generating an array of structurally distinct lipids such as mycolic acid and multi-methyl branched fatty acids [110]. These complex lipids are known to determine the virulence and pathogenesis of *Mtb* [111].

The task of biotinylation of target proteins is performed by biotin protein ligase enzymes (BPL) encoded by *birA* [109,112]. The process of biotin biosynthesis is discussed in detail elsewhere [100,112,113]. Briefly, the biogenesis can be differentiated into two stages: (i) synthesis of precursor compound pimelate thioester (pimeloyl-ACP); and (ii) conversion of pimeloyl-ACP to biotin requires the activities of four conserved enzymes, namely 7-keto-8-aminopelargonic acid synthase (KAPAS, *bioF*), 7,8-diaminopelargonic acid synthase (DAPAS, *bioA*), dethiobiotin synthase (DTBS, *bioD*), and biotin synthase (BS, *bioB*) [109,112,113,114,115]. Lastly, the insertion of the S moiety at the C6 and C9 positions of dethiobiotin (DTB) intermediate generates a thiophane ring of biotin. BS, similar to MiaAB and LipA, is categorized as a member of the radical SAM superfamily, which utilizes SAM for radical formation [109,115,116,117]. BS purified from *E. coli*, works as a homodimer harboring a [4Fe-4S]^2+,1+^ cluster and a [2Fe-2S]^2+^ cluster [109,113]. Studies have shown that [4Fe-4S]^2+,1+^ cluster donates an electron to mediate reductive cleavage of SAM to generate 5′-deoxyadenosyl radical and methionine [109,113,116]. The radical then activates the C-H bonds at C9 and C6 of DTB by extracting a proton each; hence, two SAM moieties are required for the first half-reaction. Recent biochemical and biophysical experiments suggest that the [2Fe-2S]^2+^ cluster serves as a sacrificial S donor, following which the cluster is damaged and needs Fe-S cluster reconstitution machinery for preparing BS for another round of catalysis [113,118]. Indeed, CSD (NifS/IscS) is capable of stimulating enzymatic activity of BioB (in vitro) by providing S for its Fe-S cofactor generation, thus highlighting the role of CSD in biotin-homeostasis [119]. Under pathophysiological conditions, knockdown of the BPL encoding gene in *Mtb* resulted in rapid killing in both acute and chronic infection of mice, highlighting the importance of biotin in *Mtb* survival inside the host [120].

### 4.5. Thiamine Synthesis

Vitamin B1 (thiamine pyrophosphate (TPP)) is a cofactor for enzymes involved in the biosynthesis of branched-chain amino acids and carbohydrate metabolism [121]. TPP is formed from two precursor heterocyclic compounds, 4-amino-5-hydroxymethyl-2-methylpyrimidine pyrophosphate (HMP-PP) and 4-methyl-5-(β-hydroxymethyl) thiazole phosphate (THZ-P), which are synthesized through independent pathways [116,117]. The thiazole phosphate intermediate is generated from tyrosine, cysteine, and 1-deoxy-D-xylulose-5-phosphate, through the involvement of at least six gene products (ThiFSGH, ThiI, and ThiJ) [116,122,123,124]. Mass spectrometric data has shown that ThiS and ThiF co-purify, and ThiS gets post-translationally modified at the carboxy terminus to a thiocarboxylate (ThiS-COSH), which acts as primary S source to thiazole [125]. The IscS mutants are auxotrophic for thiamine and nicotinic acid [76]. The phenotype could be recovered by supplementation with the thiamine precursor 5-hydroxyethyl-4-methyl-thiazole, signifying the requirement of IscS as a S-donor for thiazole biosynthesis [76,121,122,126,127]. IscS procures the S from L-cysteine, which is relayed on to ThiI (a rhodanese family protein), which further aids in thiolation of activated ThiS (acyladenylated) [128]. The influence of IscS on thiamine biosynthesis could also be mediated through a Fe-S cluster containing radical SAM superfamily enzyme ThiH, which is a tyrosine lyase that acts on the last step of thiazole synthesis [122,123]. Therefore, TPP synthesis is intricately connected with CSD activity. However, thiamine auxotrophy is absent under anaerobic conditions [76]. This can be due to either: (i) a decreased requirement of TPP using enzymes during anoxia; or (ii) Fe-S cluster of ThiH can be more sensitive to O_2_/ROS-mediated damage. This remains unclear and needs further investigation.

### 4.6. Molybdopterin Synthesis

Molybdopterin (MPT) cofactor (Moco) is constituted of mono-nuclear molybdenum covalently linked to a dithiolene moiety of the tricyclic pterin backbone [129,130]. Enzymes harboring Moco can be classified into three families based on the Moco derivative inserted: (i) xanthine oxidase (XO) family; (ii) sulfite oxidase (SO) family; and (iii) dimethyl sulfoxide (DMSO) reductase family [130]. The Xanthine oxidase family performs reactions involving oxidative hydroxylation of aldehydes and aromatic heterocycles that requires C-H bond cleavage. Enzymes such as dissimilatory nitrate reductase, formate dehydrogenase, and biotin-S-oxide reductase belong to the DMSO reductase family and function in C, N, or S metabolism [131]. In most organisms studied to date, Moco synthesis is mediated by a conserved biosynthetic pathway that can be segregated into three stages: (i) generation of the cyclic pyranopterin monophosphate (cPMP); (ii) introduction of two S atoms into cPMP forming MPT; and (iii) molybdate insertion to form Moco [132]. The process initiates with the formation of cPMP from 5′-GTP mediated by the enzymes MoaA and MoaC; MoaA contains two [4Fe-4S] clusters marking an influence of Fe-S cluster availability on Moco synthesis [132]. In the consequent step, MPT synthase (MoaD-MoaE) converts cPMP to MPT, via the formation of thiocarboxylate on the C-terminus of MoaD. Here, MoaD acts as the final S donor, which, in turn, derives S from CSD [132]. In *E.* coli, IscS is demonstrated to physically interact with MoeB and MoaD (the MPT synthase) using surface plasmon resonance analysis [133]. The same study also reports the stimulation of IscS activity due to MoeB and MoaD interaction. Indeed, decreased sulfuration levels of MoaD were observed in *iscS* but not *sufS* or *csdA* deleted *E. coli* strains, demonstrating that IscS is the primary S-donating enzyme to Moco synthesis [133].

### 4.7. Hydrogen Sulfide Production

Hydrogen sulfide (H_2_S), attributed as a gasotransmitter, modulates physiological pathways such as inflammation, angiogenesis, cancer biology, and oxidative stress defense [134,135,136]. The chemical characteristic of H_2_S, along with several studies, suggest that it can influence the basal cellular redox physiology via scavenging ROS and RNI, reacting with metal centers, interact with protein cysteine thiols generating persulfide, modulating cellular respiration, and antibiotic tolerance [135,136,137]. In addition to the evolutionarily conserved transsulfuration pathway and 3-mercaptopyruvate sulfurtransferase (3-MST) enzyme, CSD catalyzes the synthesis of H_2_S and alanine under reducing conditions [138]. Moreover, a recent study in *E. coli* has shown that the cysteine desulfurase IscS and not 3-MST, is responsible for H_2_S-production under reducing and anaerobic growth conditions [138]. Indeed, the *E. coli iscS* mutant strain exhibited reduced H_2_S levels under anaerobiosis and impaired growth, which was fully restored when treated with an H_2_S donor (Na_2_S: sodium sulfide). In sum, the findings suggest that H_2_S generated by IscS activity is a function of the redox state of the bacterial cytoplasm and environmental O_2_. Another study in pathogen *Mycoplasma pneumonia* identified CSD HapE to be responsible for the generation of H_2_S, which acts as a virulence factor by potentiating hemolysis [139]. Whether a similar H_2_S-mediated defense mechanism is exploited by other human pathogens such as *Salmonella* or *Mtb* remains unanswered and needs to be examined. In sum, CSD contributes towards producing the gaseous molecule H_2_S and, thereby, likely impacts bacterial pathogenesis, stress-defense, and antibiotic response.

### 4.8. DNA Phosphorothioation

Phosphorothioate (PT) modification of the DNA sugar-phosphate backbone involves replacing nonbridging oxygen atom in phosphate-moiety by S [140]. PT-modification is governed by two systems- DndABCDE or SspABCD machineries where DndA/IscS and SspA are respective CSDs (Type I) [140,141]. Microbes have evolved this epigenetic modification to protect themselves from methylation-based Restriction Modification (RM) systems and phage-invasion [142]. However, additional functions have been recently attributed to CSD-mediated PT modification. The PT-deleted bacterial strains suffer from increased intracellular ROS accumulation and ROS-mediated damages and hypersensitive to exogenous ROS, suggesting their direct role in ROS scavenging [143,144]. Additionally, the PT modification of DNA also influences gene expression and metabolic rerouting to counter ROS [142,143,144].

Taken together, CSD represents a central enzyme that mobilizes S towards diverse physiological pathways—ranging from epigenetic regulation, gene expression, and redox-signaling to steady-state metabolic pathways, indicating the indispensability of CSDs.

## 5. A Multi-Layered Regulation System Modulates the Expression and Activity of Cysteine Desulfurase in Bacteria

### 5.1. Transcriptional Level

The CSD genes are mostly co-transcribed along with genes involved in Fe-S cluster biosynthesis. The expression of *isc* or *suf* genes is influenced by either housekeeping-signals such as Fe-S cluster levels [145], or environmental stimulus such as Fe-limitation [145], ROS, RNI [146,147], hypoxia, nutrient starvation, and inside activated macrophages [148,149,150]. In *E. coli*, the regulation of ISC and SUF is primarily mediated by the transcription factor IscR with the help of other accessory transcription factors, including OxyR (peroxide responsive regulator), Fur (ferric uptake regulator), and IHF (integration host factor) (Figure 6) [147,150]. IscR is a [2Fe-2S]^+^ containing homodimeric protein, which acts as a sensor of cellular Fe-S cluster levels and oxidative stress [151]. The *E. coli* IscR belongs to the MarA/SoxS/Rob family of transcription factors and encoded by the first gene in the operon *iscRSUA-hscBA-fdx* [151]. Depending on the [2Fe-2S]^+^ cofactor bound status, apo- or holo-IscR exhibit altered binding affinity towards *isc* promoter (P*_isc_*) region. This repression of the *isc* operon by holo-IscR requires a functional ISC system, which fulfills the Fe-S cluster demand for the cell. Growth under aerobic conditions puts a high demand for Fe-S clusters due to the need for replenishing apo-protein substrates, which competes with IscR cluster occupancy [146,150,151,152]. Due to the relatively low binding affinity of apo-IscR for [2Fe-2S]^+^, it tends to remain in the apo-form during high Fe-S demand, leading to alleviating the repression of *isc* operon [150,151,153]. Whereas during anaerobiosis, the need for Fe-S enzyme maturation by the ISC system is satisfied, leading to a higher proportion of holo-IscR, which results in the repression of the *isc* operon [150,153].

The *suf* operon in most Gram-negative bacteria is shown to be functional under conditions of high demands for Fe-S cluster biogenesis and repair (e.g., iron starvation and oxidative stress) [146,150]. The *suf* operon (*sufABCDSE*) in *E. coli* contains an upstream regulatory region comprising of three *cis*-acting oxidant responsive elements (ORE-I, II, III) that serve as the interacting sites for factors like OxyR, IHF, IscR, and Fur [150,153]. In the presence of an oxidizing environment, the oxidized-OxyR binds to the ORE-I site facilitating interaction with RNA polymerase through a loop formed by IHF bound to the ORE-II site and induces expression [153]. Expression of *suf* is also controlled by Fur, which senses cellular-Fe^2+^ levels. *suf* expression is maintained at minimal levels by Fur mediated repression in reducing or Fe-abundant conditions by binding to ORE-III [153]. Interestingly, apo-IscR also binds to the ORE-III site and activates the expression of *suf* in a mutually exclusive manner to Fur binding [153]. This implies that under Fe^2+^-replete conditions, the SUF system will be repressed by dual control of Fur and IscR.

Unlike Gram-negative bacteria, Gram-positive bacteria generally encode for *suf* operon [45,150]. This suggests the absence of IscR mediated regulation and the involvement of alternate regulatory mechanisms. Cyanobacteria (e.g., *Synechocystis* sp.) and *Mtb* harbors a SUF system that lacks *sufA* but contains a regulatory gene *sufR*. Similar to *iscR*, *sufR* encodes for a Fe-S cluster containing transcriptional repressor of the *suf* operon [52,153,154,155,156]. *Synechocystis* sp. also shows two genes encoding for IscS-like CSD elsewhere in the genome [155]. Similarly, *Mtb* genome consists of a standalone *iscS* gene, encoding a functional CSD [52,76]. Intriguingly, this *Mtb iscS* transcript is among the 26% of *Mtb* genes that form a leaderless transcript, thus marking yet another undiscovered way of regulating CSDs [157].

### 5.2. Post-Transcriptional Level

The ISC system is post-transcriptionally regulated by a small regulatory RNA (sRNA), RyhB. RyhB expression is regulated by Fur, which remains in holo-form (Fe^2+^) under Fe replete conditions. Holo-Fur binds upstream of the *ryhB* sequence and inhibits its expression [158,159]. In contrast, under iron-limiting conditions, RyhB is transcribed and binds to the polycistronic *i**scRSUA* mRNA at the *iscS* Shine-Dalgarno sequence. This RhyB-mRNA complex further recruits the ribonuclease RNaseE to initiate degradation of the downstream *iscSUA* transcript, leaving the IscR-encoding part intact [158,159]. This causes unbalanced translation to form only apo-IscR, which then activates the *suf* operon. Therefore, the sRNA RyhB does not interact with the *suf* operon directly but regulates the expression of *suf* genes by modulating the expression of IscR (Figure 6) [150,159].

### 5.3. Post-Translational/Enzymatic Level

Multiple factors regulate CSD enzymatic activity. The activity of SufS is increased upon interaction with the SufBCD complex in *E. coli* and SufU in *B. subtilis* and *Enterococcus faecalis* [36,160,161]. In eukaryotes, a small accessory protein, Frataxin (Yfh1), interacts with CSD Nfs1 and scaffold protein Isu1 to aid Fe-S cluster biogenesis [162]. Frataxin is essential in humans, and its deletion impairs Fe-S homeostasis and causes neurodegenerative disease Friedreich’s ataxia [162]. Frataxin homolog in bacteria, CyaY can bind Fe^2+^ or Fe^3+^ (iron chaperone) and can donate Fe for Fe-S cluster biogenesis in vitro [163,164,165]. However, *cyaY* is dispensable in prokaryotes during normal growth conditions [166]. Interestingly, CyaY has been shown to counteract oxidative stress and functions to repair Fe-S-containing enzymes such as Respiratory Complex I (NdhI), and Acn [167,168]. In vitro studies have demonstrated that [2Fe-2S]-Ferredoxin physically interacts with CSD and either donates an electron for Fe-S assembly or Fe^2+^ donor [169]. In addition to this, CyaY also competes with Ferredoxin and IscU as an iron-dependent inhibitor of CSD activity in vitro [169]. Therefore, the role of frataxin-homologs in prokaryotes is incompletely understood and needs to be further investigated. For more details, readers are encouraged to refer to the published article [10].

CSD activity is also redox-regulated and mediated by the thioredoxin system (TrxA (Thioredoxin)/ TrxR (Thioredoxin reductase)) [170]. In vitro biochemical assays demonstrated that the reactions of *B. subtilis* SufS:SufU sulfurtransferase and TrxR to be coupled, involving NADPH-dependent reduction of SufU-persulfide. Inactivation or deletion of the Trx system impaired the activity of Fe-S cluster containing enzymes, suggesting a link between cellular redox buffer-thioredoxin and CSD activity.

## 6. Cysteine Desulfurase Maintain Intracellular Redox Homeostasis and Impart Oxidative-Stress Defense in Diverse Bacterial Species

### 6.1. Role in Sustaining Intracellular Redox Balance

CSD influences the flux of Fe (Fe^2+^, Fe^3+^) and S (S^2−^) into diverse cellular pathways through Fe-S biogenesis. Fe-S clusters are redox-active moieties that are evidently both the cause and target of oxidative stress-mediated damage [171]. ROS (O_2_^•−^ and H_2_O_2_) oxidize Fe-S clusters, especially from dehydratases, leading to the leaching of Fe and disintegration of Fe-S clusters [72]. This reaction has two primary consequences: First, the increase in labile Fe^2+^ pool feeds into the Fenton reaction to generate highly reactive ^•^OH radicals, which induce oxidative damage to DNA, lipids, and proteins [172]. Second, the disintegration of Fe-S clusters renders the enzymes inactive. Indeed, mutants for CSD lead to decreased carbon catabolism through inhibition of TCA cycle enzyme Acn, decreased respiration, and ATP production, all of which are essential for cellular metabolism and growth [51,173]. Additionally, disruption of the electron transport chain (ETC) can potentially cause electron leaks to O_2_ generating O_2_^•−^ [174,175]. Therefore, constant maintenance of Fe-S cluster biogenesis and repair by CSD is essential in decreasing ETC-generated ROS and thwarting ROS-mediated damages.

In addition to ETC, intracellular ROS is generated when electrons are adventitiously transferred to O_2_ via flavoenzymes, transition-metal centers, and respiratory quinone-containing enzymes [176]. Cells keep this basal-ROS below toxic levels via antioxidant enzymes, regulating metabolic fluxes and repair proteins. Interestingly, lipoic acid, one of the end-products of CSD activity, participates in redox metabolism by functioning as a redox couple. Lipoate (oxidized)/dihydrolipoate (reduced) can quench an array of toxic free radicals, which include ^•^OH, peroxyl radical, O_2_^•−^, and singlet oxygen species [98,177,178]. In addition to acting directly as free radical quenchers, dihydrolipoate works synergistically with multiple antioxidant systems such as vitamin C, glutathione, coenzyme Q10, and vitamin E to maintain ambient redox-homeostasis [179].

### 6.2. Function of CSD in Neutralizing Exogenous Redox Stress

Direct implications of CSD in defense against oxidative stress have been documented in several pathogenic species. The intracellular pathogen *Mtb*, is exposed to various oxidative and nitrosative insults within alveolar macrophages [180]. Interestingly, *iscS* deleted *Mtb* displayed hypersensitivity towards H_2_O_2,_ and the overexpressing strain was more resistant to oxidative stress implying its protective role [51]. *Mtb* IscS directly interacted to coordinate the activity of essential Fe-S cluster containing proteins (e.g., Acn and SdhB) [51]. In addition, *Mtb* IscS assembles the Fe-S cluster of the redox-active transcription factor, WhiB3, in vitro [71]. WhiB3 owing to its redox-sensitive Fe-S cluster, senses host-derived signals such as ROS, RNI, and low pH to assist *Mtb* in maintaining redox balance and persist in vivo [71,181]. However, in addition to IscS, *Mtb* harbors a second putative CSD SufS (Rv1464) in the *suf* operon (*rv1460–rv1466*). It is unknown how much SufS contributes to the total *Mtb*-CSD activity under normal growth conditions and stress. Therefore, dissecting the contribution of SufS in *Mtb* pathogenesis remains to be elucidated.

A similar role of CSD in defense against antibiotic-induced ROS has been documented in gut-pathogens. Antibiotic administration increases gut O_2_ levels, unlike the hypoxic state that prevails typically in the gut epithelium [182]. During infection, *Clostridium difficile* IscS2 impart resistance to toxic O_2_ levels, thereby allowing successful murine gut invasion [21]. The loss of *iscS2* in *C. difficile* led to growth and sporulation defects in addition to being sensitive to the 2% O_2_ in the gut [21]. *Helicobacter pylori*, another gastric pathogen, poses a singular NIF-system for its Fe-S cluster biosynthesis [183]. Transcriptional studies revealed the *nifS-nifU* operon to be highly upregulated in the presence of 12% O_2_ or when *H. pylori* were cultured in Fe-rich conditions, suggesting the role of Fe-S biogenesis and repair as means to limit O_2_-toxicity [183]. Comparable data are recapitulated in parasitic protozoan—*Leishmania donovani*. This pathogen faces oxidative stress within phagocytes, during heme-degradation and complement lysis in blood meal [24]. *L. donovani* circumvents damage of its labile Fe-S clusters in the presence of ROS generators menadione and amphotericin-B (a clinically- relevant drug against visceral leishmaniasis) by upregulating *Ld-iscS* [24].

*E. coli* contains three Fe-S cluster biogenesis and repair machinery, namely SUF, ISC, and CSD systems. The *E. coli* SUF S-transfer system exhibits higher CSD activity under physiological substrate conditions and upon exposure to H_2_O_2_ compared to the analogous ISC system [184]. Additionally, the catalytic cysteine (C_328_) of IscS was more susceptible to dead-end oxidative modifications than that of SufS (C_364_) [184] in the presence of H_2_O_2_. On this basis, the SUF-system is more suited for Fe-S cluster biogenesis under exogenous redox stress [89,107,155]. *E. coli* IscS is also involved in redox-dependent H_2_S biogenesis, where reducing conditions stimulate H_2_S production [138]. The H_2_S gas is known to protect diverse bacteria from oxidative stress and antibiotics [185,186]. It maintains growth and bioenergetics under hypoxia as well [137,138].

Nitrogen-fixing cyanobacteria *Anabaena* encodes four CSD ORFs (*all1457, alr2495, alr3088,* and *alr2505*). Alr 2495 (annotated as SufS) and accessory protein AsaE on overexpression, reduced intracellular ROS in the presence of H_2_O_2_ and improved survival [22]. O_2_-induced expression of SufS2 was also important for survival in *Agrobacterium tumefaciens* [23]. Toxic potassium tellurite (K_2_TeO_3_) has a deleterious effect on bacteria due to the oxidation of cellular thiols such as glutathione (GSH) [187]. *Geobacillus stearothermophilus* IscS, when expressed in *E. coli* showed cellular resistance to K_2_TeO_3_. A similar observation was made in *Staphylococcus aureus, E. coli,* and *Rhodobacter sphaeroides* [188,189]. Altogether, these studies indicate the importance and association of CSD in controlling intracellular ROS levels, which confers protection against environmental and host-induced redox stress.

## 7. Cysteine Desulfurase Is a Potential Drug Target-Candidate

Recent studies suggest that CSD could be an ideal drug target for infectious pathogens. Prokaryotic SUF machinery is significantly distinct from their eukaryotic counterparts [44,190], and targeting it could affect a multitude of essential functions dependent on Fe-S cluster biogenesis [4,5]. Moreover, differential expression of SUF and IscU has been shown to increase resistance against aminoglycosides, suggesting that antibiotic efficacy is linked to CSD-driven Fe-S homeostasis [20,173,191].

Studies by Kohanski et al., provided direct evidence linking the efficacy of three diverse bactericidal antibiotics—ampicillin (β-lactam), kanamycin (aminoglycoside), and norfloxacin (fluoroquinolone)—with IscS in *E. coli* [20]. These antibiotics trigger ROS-generation as a secondary mechanism to induce killing. Interestingly, *E. coli*-*iscS* deleted strain accumulated less ^•^OH radical upon antibiotic treatment and exhibited reduced lethality [20]. This led the authors to propose that Fe released from the ROS-damaged Fe-S clusters stimulate the generation of Fenton-mediated ^•^OH radical during antibiotic exposure. In *E. coli iscS* mutant, the depleted Fe-S cluster pools impair Fenton reaction and contribute to reduced susceptibility to antibiotics. However, Ezraty et al., found that *E. coli*
*iscS* mutant is resistant to aminoglycoside due to defects in Fe-S cluster maturation of Ndh1/2 and complex 2 (Sdh), which led to aberrant proton-motive force (PMF) and inefficient uptake of aminoglycoside [173]. Interestingly, clinically resistant-isolates derived from visceral leishmaniasis patients, which are unresponsive to Amphotericin-B treatment, showed decreased CSD-expression and activity, along with reduced ROS levels than the sensitive strain [24]. Taken together, these studies suggest that active CSD potentiates antibiotics efficacy and CSD inhibition induces antibiotic resistance. Although CSD-activity appears to increase the bactericidal efficacy of antibiotics, it is important to note that, in multiple studies, targeting this enzyme has better treatment outcomes probably by disturbing multiple essential metabolic pathways where it serves as the S-donor (Figure 7).

Several inhibitors of CSDs have been previously reported. Firstly, cysteine analogs, L-allylglycine, and L-vinylglycine, irreversibly inactivated NifS by forming a ɣ-methylcystathionyl or cystathionyl residue, respectively, through nucleophilic attack by an active site cysteinyl residue on the corresponding analog-pyridoxal phosphate adduct [63,192]. Additionally, alkylating agents such as 1,5-I-AEDANS significantly inhibited IscS and NifS activity [192]. As CSDs are PLP-dependent enzymes, molecules that form stable adducts with PLP cofactor can irreversibly inhibit CSDs [63]. Charan et al. demonstrated that PLP-fold type1 (aspartate aminotransferase family) targeting molecule, D-cycloserine (DCS), forms an adduct with the PLP-cofactor bound to *Plasmodium falciparum* PfSufS (PfSufS-PLP-DCS; 3-hydroxyisoxazole-pyridoxamine derivative), thereby inhibiting PfSufS activity and impeding the growth of *the parasite* [193]. The 50% inhibitory concentration (IC_50_) of DCS against *P. falciparum* is ∼29 μM. Interestingly, DCS also displays inhibitory activity against *Mtb* and is in clinical use as a second-line anti-TB drug [193,194,195]. However, future experimentations are needed to examine if DCS inhibits *Mtb* SufS. In sum, the inhibition of SufS by DCS provides a proof of concept for designing inhibitors targeting the PLP moiety of this primary enzyme of S-mobilization and Fe-S biogenesis pathway.

SufS delivers S from cysteine to SufBCD complex during the generation of Fe-S clusters [196]. In a significant discovery, Choby et al. identified small-molecule VU0038882 (’882) as a first-in-class inhibitor of Fe-S cluster machinery in *S. aureus* [197]. ’882 directly interacts with SufC in complex with SufB and SufD to inhibit Fe-S assembly, thereby disrupting CoA homeostasis and multiple other metabolic pathways [197]. Indeed, ’882 is extremely toxic to both aerobic (IC_50_ ∼5 μM) and anaerobic (IC_50_ ~162 μM) *S. aureus* [197], and a ’882-derivative molecule significantly reduced the burden of *S. aureus* in a mice-infection model [198]. In sum, the deleterious and pleiotropic effects of ’882 are mediated by defective Fe-S cluster assembly, demonstrating ISC/SUF/NIF system as an effective and novel drug target. These findings present the opportunity to develop the ISC/SUF machinery inhibitors as a potential novel therapeutic to control AMR.

## 8. Conclusions and Future Directions

A plethora of literature over the past few decades stands testament to the role of S-relay and CSDs in pathogenic metabolism, virulence, and stress response [4,5,6]. Cysteine desulfuration remains paramount in incorporating S in coenzymes and cofactors (Fe-S clusters, lipoic acid, biotin, etc.) on which cellular physiology relies. Our review highlights this aspect and describes how central metabolism, transcriptional regulation, DNA, and protein biosynthesis are conjoined by CSDs. Corresponding to the physiological importance, CSDs also influence the pathogenesis of disease-causing organisms. This is emphasized by how pathogenic-CSD manipulates cellular physiology to alleviate drug-induced toxicity and how they evade various host-derived oxidative stresses [20,21,24,51,199]. Moreover, multiple studies wherein “paralogs” of CSD are mutated simultaneously have rendered the organism non-viable. The multi-functionality of CSDs corroborates its essentiality. Therefore, CSDs should be extensively explored as a potential drug target that may augment existing drug regimens.

## Figures and Tables

**Figure 1 antioxidants-10-00997-f001:**
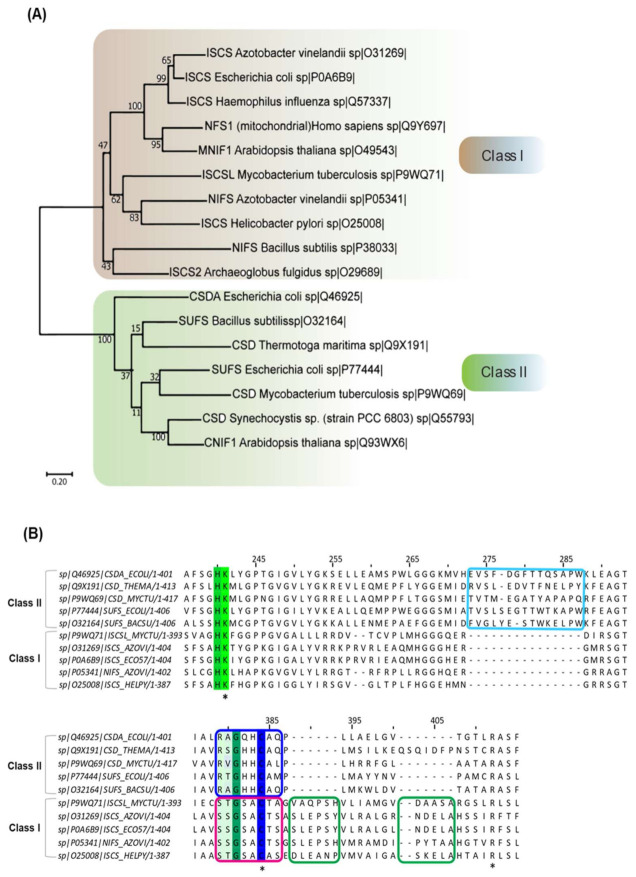
Phylogenetic analysis of CSD enzymes of organisms from different domains. (**A**) Maximum likelihood phylogenetic reconstruction of 17 CSD homologs from organisms spanning throughout three domains: Eukarya, Bacteria, and Archaea. Evolutionary analysis performed in MEGA X [42]. The CSD homologs are seen to cluster into two main branches, which can be determined as Class I and II. This tree has a maximum log-likelihood of −13398.96. The scale bar shows genetic distance. Bootstrap values (500 replicates) shown next to the branches. (**B**) Multiple sequence alignment of different bacterial CSD homologs. CSDs are indicated as UniProt ID. The green-highlighted lysine (K)-residue is the highly conserved PLP-coordinating residue, while the cyan box highlights the sequence insertion acquired by the Class II CSDs. The green boxes focus on the sequence insertion inherent to the Class I CSDs, whereas the blue and pink box features the differences in the sequences adjacent to the active site cysteine. Conserved lysine, cysteine and arginine residue are indicated by “*”.

**Figure 2 antioxidants-10-00997-f002:**
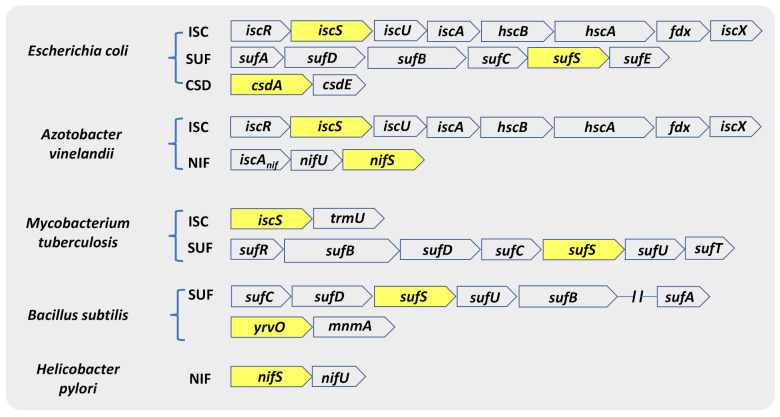
Genetic arrangement of ORFs coding for different classes of CSD. CSD genes in various organisms mostly remain associated with the genetic loci encoding Fe-S cluster biogenesis machinery (ISC, SUF, and NIF). The genetic map also shows that different bacterial species have evolved to harbor different combinations of Fe-S cluster biogenesis machinery. Genes indicated inside yellow boxes are the respective CSD of that operon.

**Figure 3 antioxidants-10-00997-f003:**
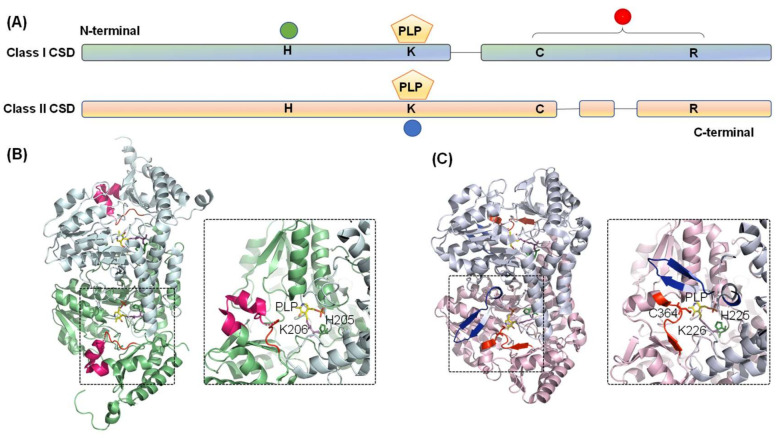
Diagram representing the domain architecture of CSD from Classes I and II. (**A**) The green circle indicates conserved histidine residue involved in acid-base catalysis and interaction with PLP-L-cysteine. Conserved cysteine and arginine residues are in the catalytic loop region, as indicated by the red circle. PLP is pyridoxal-5′-phosphate is covalently bound to conserved lysine residue (blue circle). Comparison of the three-dimensional structures of the (**B**) Class I (*E. coli* IscS, PDB ID- 3LVM) and (**C**) Class II (*E. coli* SufS, PDB ID- 6UY5) CSD enzymes highlights the differences between the two classes. Conserved histidine, lysine, and catalytic cysteine residues are shown as stick representation in forest green, magenta, and red. The PLP moiety is shown in yellow. Identifiers I–III are highlighted in red, blue, and hot pink, respectively.

**Figure 4 antioxidants-10-00997-f004:**
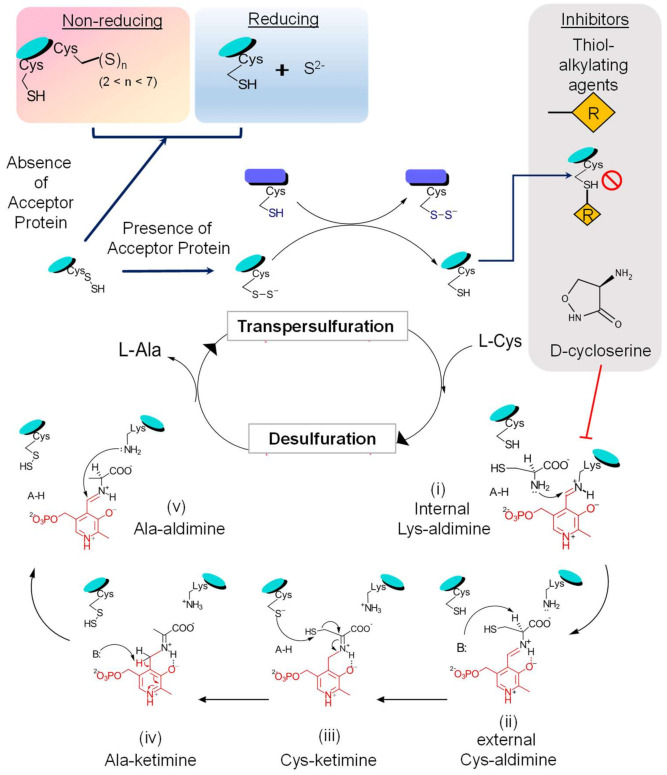
Schematic representation of the CSD reaction mechanism. The enzymatic activity of CSD can be divided into two phases: firstly, desulfuration of substrate L-cysteine and formation of the enzyme-bound persulfide intermediate, followed by the transfer of persulfide to its acceptor proteins, transpersulfuration. During the first phase of the reaction, the covalently bound cofactor-PLP acts as an electron sink and aids in the nucleophilic attack and the concomitant S-abstraction. This reaction proceeds via the formation of several reactive intermediate steps. The outcome of transpersulfuration can depend on the adjacent redox environment and the availability of S-acceptor proteins. On interaction with S-acceptor proteins, there is a transfer of the CSD-bound persulfide to the reactive thiol of an active site cysteine residue in the acceptor protein. However, in the absence of acceptors, the persulfide can get released as S^2−^/H_2_S under reducing conditions or form polysulfide species S_n (2 < n < 7)_ on an adjacent free thiol under non-reducing conditions. Furthermore, the activity of CSD can be inhibited by thiol-alkylating agents (such as N-ethylmaleimide, iodoacetamide) by irreversibly modifying the thiol group of the active cysteine residue. CSDs, being PLP-dependent enzymes, are inhibited by molecules that covalently interact with the PLP-moiety forming irreversible protein-inhibitor complexes (for example, D- or L-cycloserine).

**Figure 5 antioxidants-10-00997-f005:**
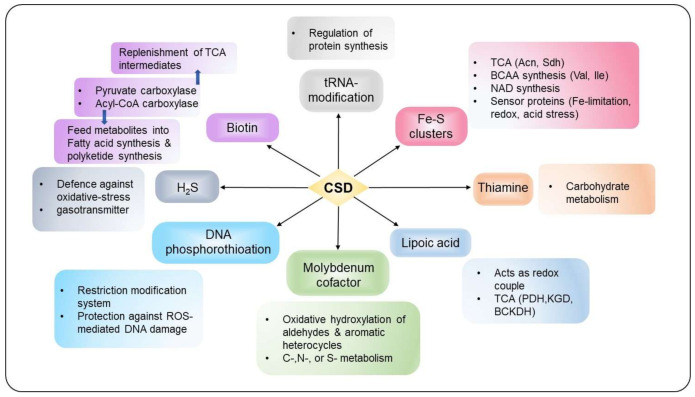
The interconnected network of S-transfer and incorporation into different thio-cofactors and tRNA. CSDs act as a pivotal point between diverse physiological functions, including cellular metabolism, stress response and protection, regulation of gene expression, and virulence.

**Figure 6 antioxidants-10-00997-f006:**
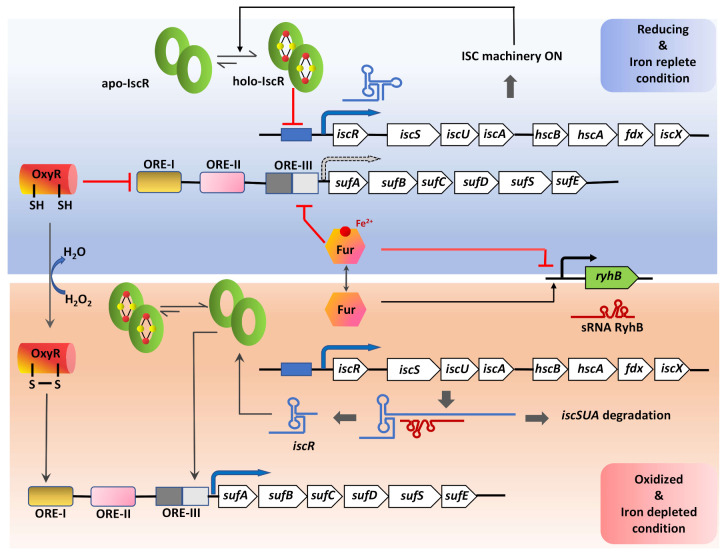
A model for the multistep regulation of Fe-S cluster biogenesis. The expression of the genes involved in Fe-S cluster biogenesis and repair are fine-tuned in accordance with the cellular requirement both under normal growth and stress conditions. This effective regulation is maintained via the role of multiple redox-sensitive transcription factors: IscR, known to regulate both the *isc* and *suf* operons with its Fe-S cluster bound (holo-) and unbound (apo-) forms; OxyR, a global regulator that senses H_2_O_2_; Fur, which senses the availability of iron. Transcript level regulation of the housekeeping ISC machinery is executed by the small non-coding RNA, RhyB, that itself is under the control of the iron-deplete/ replete form of the Fur transcription factor. Under iron-replete conditions, holo-Fur inhibits the transcription of *rhyB*, but during iron starvation, apo-Fur increases and activates transcription of *rhyB* that binds to the *isc* transcript at the *iscS* Shine-Dalgarno locus leading to its degradation, leaving the *iscR* transcript intact due to the stability imparted by its secondary structure. This, in turn, increases apo-IscR, which then binds to the ORE-III sequence upstream of the *suf* operon. Arrow with black dotted border indicates reduced expression of the *suf* operon under reducing and iron-replete conditions.

**Figure 7 antioxidants-10-00997-f007:**
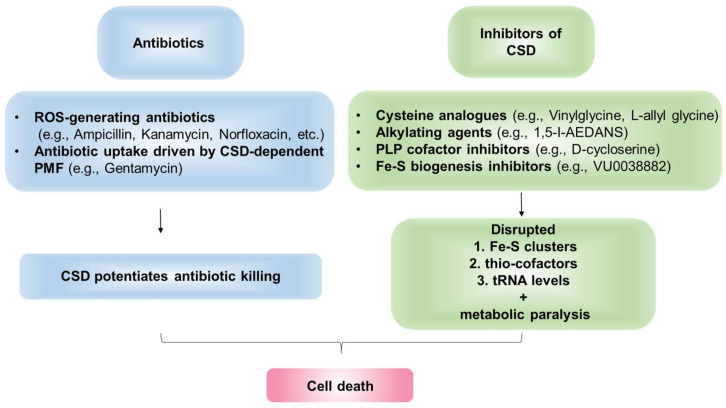
CSD as a drug target. CSDs mediate cellular death via potentiating the effect of existing antibiotics. Additionally, CSDs can be targeted by specific inhibitors to disrupt cellular physiology.

**Table 1 antioxidants-10-00997-t001:** Classification of CSDs based on primary amino acid sequences.

Enzyme	Organism	Class	Identifier I ^a^	Identifier II ^b^	Identifier III ^c^	
NifS	*Azotobacter vinelandii*	I	SSGSACTS		Insertion near conserved cysteine	Dimer
IscS	*Azotobacter vinelandii*	I	SSGSACTS		Insertion near conserved cysteine	
IscS	*Helicobacter pylori*	I	STGSACAS		Insertion near conserved cysteine	
IscS	*Escherichia coli*	I	SSGSACTS		Insertion near conserved cysteine	
SufS/CsdB	*Escherichia coli*	II	RTGHHCA	Insertion near conserved lysine		
CsdA	*Escherichia coli*	II	RAGQHCA	Insertion near conserved lysine		
SufS	*Bacillus subtilis*	II	RAGHHCA	Insertion near conserved lysine		
SufS/CSD	*Mycobacterium tuberculosis*	II	RVGHHCA	Insertion near conserved lysine		
IscS	*Mycobacterium tuberculosis*	I	STGSACTA		Insertion near conserved cysteine	
CSD	*Thermotoga maritima*	II	RSGHHCA	Insertion near conserved lysine		
C-DES *	*Synechocystis* PCC 6714					Monomer

a, b, c: CSDs from different organisms can be classified into two main categories based on their sequence identifiers: (**I**) patch of amino acid sequence neighboring the active site cysteine; (**II**) presence/absence of amino acid-stretch insertion near the conserved lysine residue coordinating PLP; and (**III**) 12-amino acid-residue insertion adjacent to the active site cysteine. * C-DES is a novel PLP-dependent L-cyst(e)ine C-S-lyase, which converts cysteine to pyruvate, ammonia, and sulfide.

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
