# Peer review of "The Multifaceted Bacterial Cysteine Desulfurases: From Metabolism to Pathogenesis"

_antioxidants, 2021, doi:10.3390/antiox10070997_

Round 1

Reviewer 1 Report

This a finely written and cohomprehensive review, covering a topic of significant interest to Antioxidant readers. Hence, it fits with the aim and scope of the journal. However, minor corrections are required to be published in Antioxidant.

  1. A figure comparing the 3D structures of Class I and II CSD is necessary
  2. the authors should carefully correct the errors in font size and type
  3. the authors should edit the reference numbering in the manuscript. i.e. Page 1, in line 42 [4]-[6] should replaced with [4-6] and so on.
  4. some typos need to be corrected

Author Response

  1. A figure comparing the 3D structures of Class I and II CSD is necessary: This information has now been added as Figures 3B and 3C, with an appropriate mention in the main text.
  2. the authors should carefully correct the errors in font size and type: The whole manuscript was thoroughly read and all errors in font size and type were corrected, both in the text as well as images.
  3. the authors should edit the reference numbering in the manuscript. i.e. Page 1, in line 42 [4]-[6] should be replaced with [4-6] and so on. : Thank you for pointing this out. We have now reformated all the references as suggested.
  4.     some typos need to be corrected: We have corrected all typing mistakes.

Reviewer 2 Report

The review by Das et al is well written and the topic is interesting.

I have some suggestions:

  • pay attention to punctuation. For example dots sometimes are missing (lanes 28, 273, 328, 352, 747) whereas in other places are too much (lanes 505, 601). Also some commas are incorrect (lanes 53, 97, 172, 174, 175, 358, 414, 507, 667).
  • Correct the superscript or the subscript in molecules ( lanes 12, 548, 623, 625, 631, 635).
  • The name of bacterial species should be entire the first time cited (lane 43, 95, 661, 687).

Typo errors in lanes 38 (dependant), 155, 298 (quonolinic).

Please correct lanes 425 (BPL encoding gene), 666 (remove “that”), 686 (on overexpression?).

In the paragraph 5.2 the figure 6 should be mentioned.

An abbreviations list would be useful.

Author Response

I have some suggestions:

  • pay attention to punctuation. For example, dots sometimes are missing (lanes 28, 273, 328, 352, 747) whereas in other places are too much (lanes 505, 601). Also some commas are incorrect (lanes 53, 97, 172, 174, 175, 358, 414, 507, 667).
  • Correct the superscript or the subscript in molecules ( lanes 12, 548, 623, 625, 631, 635).
  • The name of bacterial species should be entirely the first time cited (lanes 43, 95, 661, 687).
  • Typo errors in lanes 38 (dependant), 155, 298 (quonolinic).
  • Please correct lanes 425 (BPL encoding gene), 666 (remove “that”), 686 (on overexpression?).
  • In paragraph 5.2 figure 6 should be mentioned.

We have extensively gone through the manuscript, carefully read all the suggestions, and corrected all the above-mentioned errors.

An abbreviations list would be useful: We have now added a list of abbreviations used in the manuscript, before the acknowledgment section.
